# 'Candidatus Pseudomonas auctus' sp. nov. JDE115 isolated from nodules on soybean (Glycines max)

Md Sahadat Ali[1‡], Fatima Tuz Zohora Mony[1], Michael Evans[1], Steven Rideout[1], David Haak[1], Paulo Vieira[2], Jonathan D. Eisenback[1‡*]

1 School of Plant and Environmental Sciences, Virginia Tech, Blacksburg, Virginia, United States of America, 2 Mycology and Nematology Genetic Diversity and Biology Laboratory, United States Department of Agriculture, Agricultural Research Service, Beltsville, Maryland, United States of America

☯ These authors contributed equally to this work.
‡ These authors are joint senior authors on this work.
* jon@vt.edu

## Abstract

A Gram-negative, facultative anaerobic, rod-shaped, motile with peritrichous flagella, fluorescent bacterium, designated 'Candidatus Pseudomonas auctus' sp. nov. JDE115, was isolated from soybean root nodules in Virginia and characterized using a comprehensive integrative methodology. Growth of JDE115 occurred with 0–5.0% (w/v) NaCl (optimum 1%), at pH 6.0–10.0 (optimum pH 7.0), and at 10–40°C (optimum 28°C) in LB broth. Phylogenetic analyses based on the 16S rRNA gene placed the isolate as a member of a novel species within the genus Pseudomonas. Phylogenetic analyses based on whole-genome sequences, 16S rRNA, showed JDE115 having the highest similarity to Pseudomonas glycinae MS586. Average Nucleotide Identity (ANI) analysis also revealed the highest similarity of JDE115 to Pseudomonas glycinae MS586 (94.59%), which is below the 95% threshold for species delineation. Genome-to-genome distance analysis (GGDC, Formula 2) showed a maximum value of 57.10% with the same strain, far below the 70% cutoff. The primary isoprenoid quinone detected in JDE115 was ubiquinone-9 (Q-9) and the DNA G + C content was 60.68 mol%. The whole-cell fatty acid profile was dominated by C16:0, C17:0 cyclo, and the summed features 3 (C16:1ω7c and/or C16:1ω6c) and 8 (C18:1ω7c and/or C18:1ω6c). Additional fatty acids detected included 12:0, 14:0, and 18:0. Based on these phenotypic, chemotaxonomic, and phylogenetic data, strain JDE115 is proposed to represent a new species in the genus Pseudomonas, for which the name 'Candidatus Pseudomonas auctus' sp. nov. is proposed.

**Data availability statement:** All relevant data are within the manuscript and its Supporting Information files.

**Funding:** Virginia Tech Intellectual Property LAUNCH Proof-of-Concept Grant The School of Plant and Environmental Sciences, Virginia Tech research associate and tuition. The funders had no role in study design, data collection and analysis, decision to publish, or preparation of the manuscript.

**Competing interests:** The authors have declared that no competing interests exist.

## Introduction

The genus *Pseudomonas* was first proposed by Migula in 1894 and is known for its remarkable versatility, metabolic diversity, and environmental ubiquity [1,2]. Members of this genus are Gram-negative [3,4], rod-shaped bacteria [5] that are non-spore-forming [6] and motile due to polar flagella [6,7]. They are catalase [8,9] and oxidase-positive [10], and their ability to thrive in diverse habitats, including plants, soil, animals, and water, highlights their adaptability and ecological significance [11–16]. Currently, more than 200 species of *Pseudomonas* are listed in the bacterial names withstanding in nomenclature (https://www.bacterio.net). Many species within this genus exhibit plant growth-promoting [17–20] and biocontrol [3,4,14,17,21] properties, making them vital in agriculture for sustainable crop production and disease management. These bacteria contribute to plant health through mechanisms such as phosphate solubilization [22–27], siderophore production [28–32], and the suppression of plant pathogens via direct antagonism or secondary metabolite production [33–36].

However, accurate identification and classification of bacterial species, especially within the diverse genus *Pseudomonas*, require a multifaceted approach [37–40]. Among the most robust methods for delineating new species are genomic and phylogenetic analyses [37,41–43]. Average Nucleotide Identity (ANI) is widely used for genomic comparisons, with a threshold of 95% often considered indicative of species delineation [44–47]. Similarly, digital DNA-DNA Hybridization (dDDH) [48–49], which provides an in silico estimate of DNA-DNA reassociation, has a 70% cutoff for species-level distinctions [48,50]. These genome-based metrics complement traditional phylogenetic analyses, such as those based on the 16S rRNA gene, which has been a gold standard for bacterial taxonomy [51–53], and provide additional resolution for closely related taxa [54–58].

Chemotaxonomic approaches further strengthen taxonomic characterization by analyzing cellular fatty acids, respiratory quinones, and polar lipid profiles [59–61]. Fatty Acid Methyl Ester (FAME) analysis serves as a complementary or alternative method for detecting soil organisms. It is widely used to characterize microbial community structures and as a diagnostic tool for identifying specific organisms in environmental samples [62]. FAME analysis has also been instrumental in differentiating species within *Pseudomonas* [63–66]. Specific FAMEs or their ratios can indicate the presence and abundance of certain taxa, serving as biomarkers. For example, the presence of 10:0 3OH, 12:0 3OH, and 14:0 3OH fatty acids indicates a gram-negative organism, while their absence, along with LPS, suggests a gram-positive organism [67]. Respiratory quinones, such as ubiquinone Q-9, and polar lipids, including phosphatidylethanolamine and phosphatidylglycerol, also provide distinctive chemotaxonomic markers [68–70].

Phenotypic and physiological traits play a critical role in the identification and classification of *Pseudomonas* species [71,72]. These bacteria are characterized by their Gram-negative cell wall structure [4,19,73], essential for differentiating them from Gram-positive taxa. Their rod-shaped morphology and the presence of one or more polar flagella confer motility, enabling them to colonize diverse environments

efficiently [74–76]. Non-spore-forming and catalase-positive traits are hallmark features that further delineate members of *Pseudomonas* from other genera [40]. Additionally, their oxidase-positive nature reflects their reliance on aerobic respiration, a trait that supports their adaptability in various habitats [10]. These phenotypic traits not only confirm their inclusion within *Pseudomonas* but also provide initial clues for distinguishing closely related species. By integrating these phenotypic and physiological attributes with genomic, phylogenetic, and chemotaxonomic analyses, a comprehensive framework for species identification is achieved. Thus, these comprehensive polyphasic approaches are instrumental in accurately determining the taxonomic position of 'Candidatus Pseudomonas auctus' sp. nov. JDE115.

## Materials and methods

### Organism and growth condition

*'Candidatus* P. auctus' sp. nov. JDE115 was isolated from surface-sterilized root nodules of soybean (*Glycine max* (L.) Merr.). Briefly, 0.25 g of surface-sterilized soybean nodules (P1) were homogenized and suspended in 0.9% NaCl solution. The suspension was plated onto King's B medium for bacterial isolation (P2). A single colony of the strain was cultured in LB (Luria–Bertani) lite broth at 28°C for 24 hours under continuous shaking at 200 rpm (P3). Immediately, five samples for glycerol stock (40%) were prepared and stored at −80°C (P4).

### Cell morphology and physiological tests

The morphological characteristics of strain JDE115 colonies were assessed following growth on LB lite agar plates. Gram staining was performed according to the protocol described by Bartholomew & Mittwer [77]. Cell morphology and flagellation types were examined using 120 kV FEI T12 Tecnai Spirit transmission electron microscope (TEM) with routine negative staining using Urany-Less® [78] (P5). The attachment of cells to the root surface was observed using a Jeol NeoScope® JCM 5000 scanning electron microscope (SEM) operating at 10kV after sputter coating with 90Å of gold using an Emitech® SC7620 mini sputter coater [79,80]. Motility was visualized and recorded using a Leitz® Dialux 22 bright field and phase contrast light microscope equipped with a Nikon® D300 camera to create photomicrographs and digital video [81]. Fluorescent pigment production was evaluated by growing JDE115 on King's B medium at 28°C for 24 hours. After incubation, fluorescence was examined using a UV transilluminator (Avantor) under long-wave UV light (365 nm, UV-A) in a darkened environment. Plates were placed directly on the transilluminator surface for visualization [82]. Optimal growth conditions for JDE115 were determined by measuring optical density (OD600) in LB liquid cultures [83,84]. Growth was evaluated at temperatures ranging from 4°C to 40°C in 4°C increments over 24 hours and at pH levels from 4.0 to 10.0 to determine the strain's tolerance and preferences [2,73,85].

Physiological assessments, including salt tolerance, pH range, temperature range, motility, and aerobic metabolism, alongside comprehensive biochemical tests (unpublished, Ali et al.), were conducted to characterize 'Candidatus Pseudomonas auctus' nov. sp. JDE115. Cellular fatty acid profiles were analyzed using the Sherlock 6.1 system (Microbial Identification Inc.) with the RTSBA6 library [67], providing detailed insights into the strain's fatty acids composition (P6). Biochemical properties and enzymatic activities were evaluated using Biolog® GENIII Microplates [86–89] (P7). Results were recorded after 48 hours of incubation at 28°C, yielding a robust dataset highlighting the metabolic versatility and adaptive features of JDE115.

### Phylogenetic analysis

Bacteria from glycerol stock were streaked onto Luria-Bertani agar (LBA) plates and incubated to obtain isolated colonies. A single colony was transferred into 50 mL of LB lite broth (containing 5 g/L yeast extract and 10 g/L NaCl) and incubated at 28°C for 16 hours with continuous shaking at 200 rpm. The culture reached an optical density (OD) of approximately 0.6. From this actively growing culture, 1.8 mL was used for genomic DNA extraction using the DNeasy UltraClean Microbial

Kit (Qiagen®, USA) (P8). The extracted DNA was quantified using a NanoDrop 1000 spectrophotometer and a Qubit 3.0 fluorometer (Thermo Fisher Scientific®, USA). DNA purity was further assessed by agarose gel electrophoresis (P9). A nearly full-length 16S rRNA gene was amplified using the universal primers 27F (5′-AGAGTTTGATCMTGGCTCAG-3′) and 1492R

(5′-TACGGHTACCTTGTTACGACTT-3′) through PCR [73,90,91]. PCR was performed using a Bio-Rad® Thermal Cycler, and the products were purified with a QIAquick® PCR Purification Kit (QIAGEN®) (P10). Sanger sequencing reactions were carried out by the ABI 3730XL.

Computation of the phylogenetic tree based on 16S rRNA was performed using SeaView® [92,93] with the maximum likelihood approach with the 1000 bootstrap replicates [94,95]. Sequences of type strains used in the phylogenetic analysis were downloaded from NCBI (accession numbers in Table 3).

The genomic DNA was sheared into fragments using a Covaris® S220 ultrasonicator, and sequencing libraries were prepared using the Illumina-compatible NEBNext Ultra II DNA Library Prep Kit (New England BioLabs®, USA). Sequencing was performed on the Illumina® NovaSeq PE150 platform at the NGS core facility, generating paired-end reads of 150 bp. FASTQ files were generated using Guppy (v4.4.1), and sequences with a Phred score below 7 were excluded. De novo assembly was performed using Flye (v2.9) [96], with default parameters and a genome size setting of 6.2 Mbp. Assembly completeness was confirmed using dot plots generated with Gepard [97]. The complete genome was visualized using CGView [98,99]. Genome annotation was conducted using the NCBI Prokaryotic Genome Annotation Pipeline (PGAP) and the RAST server [100]. The Whole Genome Shotgun sequence project has been deposited in GenBank under accession number PRJNA1196930.

Similarity analyses (ANI and GGDC) of the sequenced genome of strain JDE115 to other 40 genomes of the closely related *Pseudomonas* species were determined as briefed below. ANI based on pairwise comparison was calculated using the software FastANI [44]. GGDC was calculated using the web service http://ggdc.dsmz.de and using the recommended BLAST+method [49]. The GGDC results shown are based on the recommended formula 2 (sum of all identities found in HSPs divided by the overall HSP length), which is independent of the genome length and is thus robust against the use of incomplete draft genomes.

The Type (Strain) Genome Server (TYGS, https://tygs.dsmz.de) was employed for whole-genome-based taxonomic analysis, incorporating recent methodological updates [101,102]. TYGS identified closely related type strains through two complementary approaches: (1) comparison of user genomes with type strain genomes in the TYGS database using the MASH algorithm [103] to identify the ten closest type strains and (2) extraction of 16S rDNA sequences from user genomes using RNAmmer [104], followed by BLAST [105] analysis against 16S rDNA gene sequences from 22,195 type strains in the TYGS database. These methods were used to determine the closest type strain genomes based on intergenomic distances calculated using the Genome Blast Distance Phylogeny (GBDP) approach with the "coverage" algorithm and distance formula $d_5$ [49]. Pairwise genome comparisons were performed with GBDP to infer intergenomic distances and calculate digital DNA-DNA hybridization (dDDH) values using GGDC 4.0. Phylogenomic analysis was conducted by constructing a balanced minimum evolution tree using FASTME 2.1.6.1 with 100 pseudo-bootstrap replicates for branch support [106]. The tree was rooted at the midpoint [107] and visualized using PhyD3 [108]. Species clustering was based on a 70% dDDH threshold [102], while subspecies clustering used a 79% dDDH threshold, as previously described [109]. These analyses provided insights into the taxonomic positioning of the studied genome. Taxonomic and nomenclatural information was also supplemented by the List of Prokaryotic Names with Standing in Nomenclature (LPSN, https://lpsn.dsmz.de) [101].

## Chemotaxonomic analysis

The cellular fatty acid profile of JDE115, a critical chemical characteristic for bacterial identification, was analyzed to support its taxonomic classification. Fatty acids were harvested after 24 hours of incubation at 28°C on Tryptic Soy Agar

(TSA). The extraction process involved saponification, methylation, and subsequent extraction following the standardized protocols of the MIDI System (Sherlock Microbial Identification System, version 6.0B). Analysis was performed using gas chromatography (GC; 6850, Agilent Technologies®), and the fatty acids were identified utilizing the TSBA6.0 database of the Microbial Identification System [67].

## Results and discussion

### Phenotype analysis

'*Candidatus* P. auctus' sp. nov. JDE115 was observed to be gram-negative, rod-shaped (0.5 μm) (Fig 1), fluorescent (S1 Fig), aerobic or facultatively anaerobic, motile (S1 Movie) with monotrichous flagella (Fig 1). Colonies grown on LB at 28°C for 24 h are light yellow, round, smooth, convex, waxy and translucent, 4.0–5.0 mm in diameter (S2 Fig). Growth occurs in 0–4% (w/v) NaCl (optimum 1%), at pH 4.0–10.0 (optimum pH 7.0), and at 10–40°C (optimum 28 °C). Fluorescent pigments were observed when cultured for 24 hr at 28°C on King's B medium (S1 Fig). The physiological, morphological, and phenotypic characteristics in Biolog GEN III tests, which showed differentiation of strains JDE115 from other closely related *Pseudomonas* species, are listed in Table 1.

### Phylogenetic analysis

Sequence analysis of the 16S rRNA gene revealed that '*Candidatus* P. auctus' nov. sp. JDE115 shares significant sequence identity (>99%) with several *Pseudomonas* species. The closely related strains include *P. glycinae* MS586[T] (99.86%), *P. kribbensis* 46-2[T] (99.71%), *P. soyarea* JJL17[T] (99.36%), *P. koreensis* Ps 9-14[T] (99.36%), *P. reinekei* MT1[T] (99.29%), and *P. moraviensis* 1B4[T] (99.29%). Due to the high sequence similarity, the 16S rRNA gene alone cannot determine the precise taxonomic position of these closely related species [115,116]. This limitation arises from the high conservation of 16S rRNA variable regions, taxon-specific evolutionary constraints, and detection biases, which often hinder its

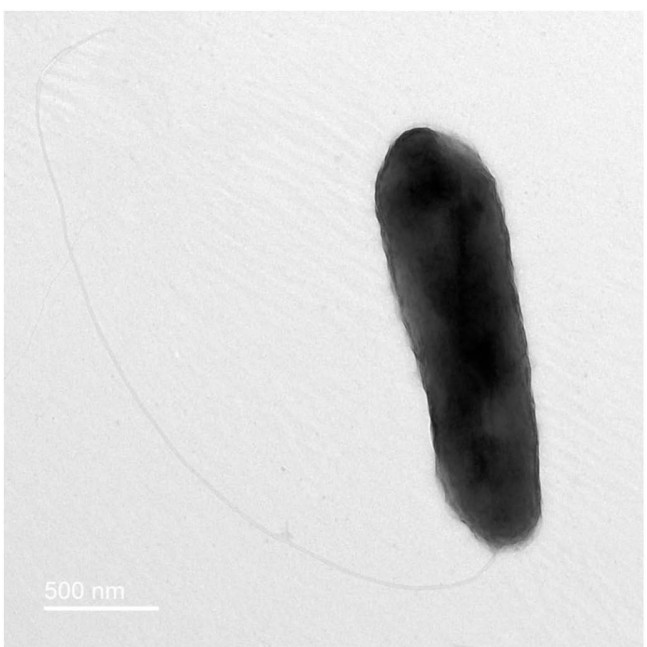

500 nm

**Fig 1. Transmission electron micrograph of '*Candidatus* Pseudomonas auctus' nov. sp. JDE115.** This rod-shaped bacterium has a monotrichous polar flagellum that is more than twice as long as its body.

**Table 1. Differentiating characteristics of 'Candidatus Pseudomonas auctus' sp. nov. strain JDE115 from other related species of Pseudomonas.**

| Characteristics | P. auctus JDE115 | P. glycinae MS586[T] | P. kribbensis 46-2[T] | P. granadensis F-278 | P. moraviensis 1B4[T] | P. koreensis Ps9-14[T] | P.baetica a390T | P. vancouverensis DhA-51[T] | P. jessenii DSM 17150[T] | P. reinekei MT1[T] |
|---|---|---|---|---|---|---|---|---|---|---|
| Flagellation | Polar, Single | Polar, multiple | Polar, multiple | Polar, two | Polar, two | Polar, multiple | ND | ND | Polar, single | ND |
| Fluorescence | | | | | | | | | | |
| Growth at 4°C | | | | | | | | ND[a] | | ND |
| Tolerance of NaCl at 5% | | | | | | | | | | |
| Nitrate reduction | | | | | | | | | | |
| Arginine dihydrolase | | | | | | | | | | |
| Hydrolysis of gelatin | | | | | | | | | | |
| Citrate utilization | | | | | | | | | | |
| Urease | | | | | | ND | | | | |
| Assimilation of l-Arabinose | | | | | | | | | | |
| N-Acetyl-d-glucosamine | | | | | | | | | | |
| Phenylacetic acid | | | | | | | | | | |
| d-Mannose | | | | | | | | | | |
| Dextrin | | | | | | | | | | |
| Tween-40 | | | | | | | | | | |
| d-Cellobiose | | | | | | | | | | |
| d-Trehalose | | | | | | | | | | |
| l-Arabinose | | | | | | | | | | |
| d-Fructose | | | | | | | | ND | | |
| d-Mannitol | | | | | | | | | | |
| d-Arabitol | | | | | | | | | | |
| l-Alanine | | | | | | | | | | |
| l-Serine | | | | | | | | | | |
| α-Ketobutyric acid | | | | | | | | | | |
| α-Ketoglutaric acid | | | | | | | | | | |
| Glucuronamide | | | | | | | | | | |
| l-Histidine | | | | | | | | | | |
| d-Serine | | | | | | | | | | |
| d-Galactose | | | | | | | | | | |
| d-Galacturonic acid | | | ND | | | ND | | | | |
| d-Glucuronic acid | | | | | | | | | | |
| Glucuronamide | | | | | ND | | | | | |
| p-Hydroxy phenylacetic acid | | | | | | | | | | |

*(Continued)*

**Table 1.** (Continued)

| Characteristics | *P. auctus* JDE115 | *P. glycinae* MS586[T] | *P. kribbensis* 46-2[T] | *P. granadensis* F-278 | *P. moraviensis* 1B4[T] | *P. koreensis* Ps9-14[T] | *P.baetica* a390T | *P. vancouverensis* DhA-51[T] | *P. jessenii* DSM 17150[T] | *P. reinekei* MT1[T] |
|---|---|---|---|---|---|---|---|---|---|---|
| Quinic acid | positive | positive | positive | positive | positive | positive | positive | negative | positive | positive |
| d-Saccharic acid | positive | positive | positive | positive | positive | positive | positive | negative | positive | positive |
| Glycyl-l-proline | positive | negative | ND | positive | positive | positive | negative | positive | positive | positive |
| L-Pyroglutamic acid | positive | positive | positive | positive | positive | ND | positive | negative | positive | positive |
| Inosine | positive | negative | positive | positive | positive | positive | positive | positive | negative | positive |
| Propionic acid | positive | positive | positive | positive | positive | positive | positive | positive | positive | weak |
| Formic acid | negative | negative | positive | negative | positive | negative | positive | negative | positive | positive |
| Acetic acid | positive | positive | positive | weak | positive | negative | positive | positive | positive | positive |
| Methyl pyruvate | positive | negative | negative | positive | positive | positive | positive | positive | positive | positive |
| GC content (%) | 60.7 | 60.5 | 60.5 | 59.9 | 60.3 | 59.1 | 58.7 | 67.2 | 62.2 | 59.1 |

Comparison of key phenotypic and biochemical characteristics of 'Candidatus Pseudomonas auctus' nov. sp. JDE115 with closely related *Pseudomonas* species, including flagellation type, fluorescence, growth at 4°C, NaCl tolerance, nitrate reduction, and various metabolic and enzymatic activities. Biochemical reactions were assessed using the Biolog GEN III system, which evaluates the utilization of 71 carbon sources and the resistance to 23 chemical agents for comprehensive bacterial identification. Data for strain 'Candidatus Pseudomonas auctus' nov. sp. JDE115 were obtained in this study. Data for other type strains were obtained from references, *P. glycinae* MS586[T] Jia et al. 2020 [73]; *P. kribbensis* 46-2[T] Chang et al. 2016 [110]; *P. granadensis* F-278,770[T] Pascual et al. 2015 [111]; *P. moraviensis* 1B4[T] Pascual et al. 2015 [111]; *P. koreensis* Ps9-14[T] Tvrzová et al. 2006 [112]; *P.baetica* a390[T] López et al. 2012 [113]; *P. vancouverensis* DhA-51[T] Cámara et al. 2007 [114]; *P. jessenii* DSM 17150[T] Cámara et al. 2007 [114]; *P. reinekei* MT1[T] Cámara et al. 2007 [114].

Abbreviations: dark green is for positive reaction; red is for negative reaction; light green is for weak reaction; and ND[a] is for no data.

ability to resolve within-genus taxonomy accurately [117]. Recent studies have also highlighted that evolutionary rigidity and horizontal gene transfer associated with the 16S rRNA gene contribute to insufficient diversification among closely related taxa, resulting in limited resolution at the species level [118]. Further whole-genome analyses and phenotypic characterizations are required to clarify the taxonomic placement and distinguish 'Candidatus P. auctus' nov. sp. JDE115 from its closely related species.

However, A phylogenetic tree based on the 16S rRNA gene sequences was constructed to assess the evolutionary relationship of 'Candidatus P. auctus' nov. sp. JDE115 with other closely related species. The analysis revealed that JDE115 shares a high sequence similarity with multiple *Pseudomonas* species, including *P. glycinae* MS586[T], *P. kribbensis* 46-2[T], and *P. soyarea* JJL17[T], forming a distinct cluster within the *Pseudomonas* genus. Despite the close relationships, JDE115 formed a unique branch with high bootstrap support, indicating its distinct taxonomic position (Fig 2).

## General genome features

The genome of 'Candidatus Pseudomonas auctus' nov. sp. JDE115 was fully sequenced and annotated, revealing a genome size of 6,183,199 base pairs (bp) with a GC content of 60.68%, the genome size of *Pseudomonas* typically range from 4.2 to 7Mba [119,120]. Besides, the GC content of JDE115 falls within the reported range of 48–68 mol% for members of the genus *Pseudomonas* [121–125]. The genome comprises 5,648 genes, of which 5,509 (97.54%) are coding genes and 70 (1.24%) are RNA genes. Additionally, the genome contains 69 pseudogenes (1.22%), representing a minor fraction of the total genetic content. The coding DNA sequence (CDS) accounted for 5,578 genes, contributing to 98.76% of total genes (Table 2). The number of coding sequences (CDS) within the genus of *Pseudomonas* exhibits substantial variability, ranging from 4,274–6,305 [126].

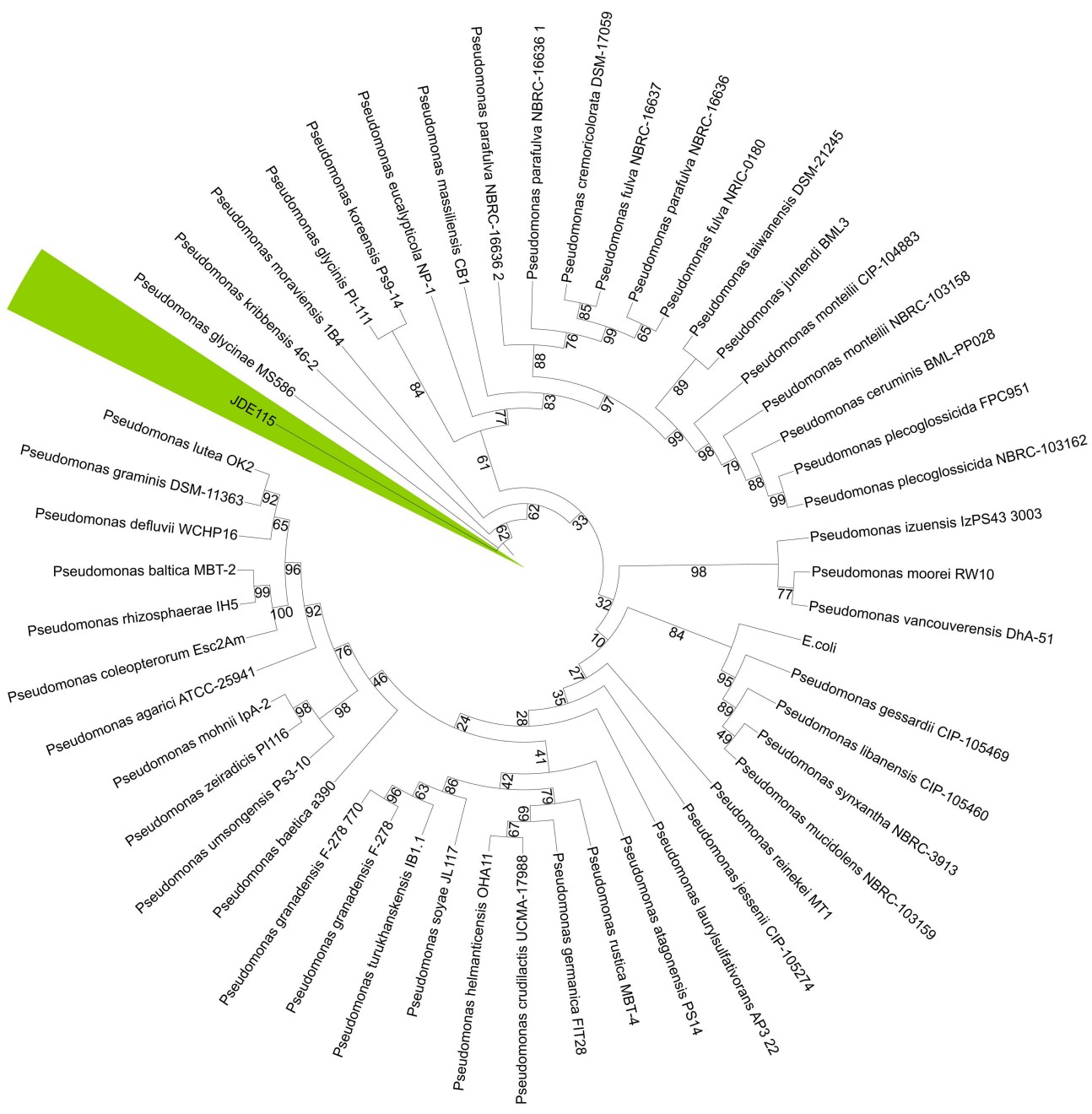

**Fig 2. Phylogenetic tree illustrating the evolutionary relationship of 'Candidatus Pseudomonas auctus' nov. sp. JDE115 with closely related species based on 16S rRNA gene sequences.** The tree was created using the Maximum Likelihood method through the PhyML engine integrated into SeaView software [92]. Bootstrap values above 50% are shown at the branch nodes, supporting the branching patterns. This analysis highlights the high sequence similarity between JDE115 and other species within the genus *Pseudomonas* while demonstrating the distinct taxonomic position of JDE115 through a unique branch with significant bootstrap support (highlighted in green).

The ANI analysis revealed the highest similarity of '*Candidatus* P. auctus' nov. sp. JDE115 with *P. glycinae* MS586, with a value of 94.59%, followed by *P. fitomatiaceae* FIT81 (94.24%) and *P. guzikowskii* IzPS32d (93.92%). However, these values fall below the commonly accepted ANI threshold of 95% for species delineation [44]. ANI comparisons

**Table 2. General taxonomic genome features of 'Candidatus Pseudomonas auctus' nov. sp. JDE115.**

| Attributes | Value | % of total |
|---|---|---|
| Genome size (bp) | 6,183,199 | 100 |
| DNA Coding (bp) | 5,513,388 | 89.17 |
| GC Content (%) | 3,752,162 | 60.68 |
| Genes (total) | 5,648 | 100 |
| CDS (total) | 5,578 | 98.76 |
| Genes (coding) | 5,509 | 97.54 |
| Genes (RNA) | 70 | 1.24 |
| Pseudo Genes | 69 | 1.22 |

Summary of the genome assembly and annotation of 'Candidatus Pseudomonas auctus' nov. sp. JDE115, including genome size, GC content, total gene count, coding sequences (CDS), RNA genes, and pseudogenes.

between genomes and type strain assemblies have corrected approximately 750 previously misidentified entries in GenBank, underscoring its critical role in taxonomic accuracy [127]. Additionally, ANI's ability to handle large datasets with high computational efficiency makes it ideal for modern microbial taxonomy, supporting robust species-level classifications even in complex datasets [128]. Similarly, digital DNA–DNA hybridization (dDDH, calculated using GGDC) values ranged from 57.10% with *P. glycinae* MS586 to as low as 21.30% (*P. wenzhouensis* A20) with more distantly related species (Table 3).

The comparative genomic analysis underscored the unique genetic identity of 'Candidatus P. auctus' nov. sp. JDE115. While ANI and dDDH analyses identified significant genomic similarities with certain *Pseudomonas* species, the values consistently remained below the established cutoffs (95% for ANI and 70% for dDDH) for species-level classification [129]. Notably, JDE115 shared substantial genomic relatedness with species like *P. glycinae* MS586 (94.59 and 57.10), *P. fitomaticsae* FIT81 (94.24 and 54.80), *P. gozinkensis* IzPS32d (93.92 and 54.00), *P. kribbensis* 46−2 (91.46% ANI and 42.50% dDDH) and *P. alokibbiensis* IzPS23 (91.74% ANI and 43.00% dDDH), yet the data unequivocally supports the novelty of JDE115 as a distinct taxonomic entity.

Average Nucleotide Identity (ANI) and dDDH values were calculated using the Genome-to-Genome Distance Calculator (GGDC). ANI values above 95% typically indicate species-level relatedness, while dDDH values above 70% suggest the same species.

A whole-genome sequence-based phylogenetic analysis was conducted for 'Candidatus P. auctus' nov. sp. JDE115 and closely related species within the genus *Pseudomonas*. The tree was inferred using the Genome BLAST Distance Phylogeny (GBDP) approach implemented in the TYGS platform. Distances were calculated from genome sequences using formula d5. The resulting phylogenetic tree illustrated that JDE115 formed a distinct branch within the genus, clustering most closely with *P. glycinae* MS586 and *P. kribbensis* KCTC 32541$^T$. Branch lengths in the tree are scaled in terms of GBDP distance formula d5, and bootstrap values from 100 replicates are provided at each node, indicating strong statistical support for the inferred relationships. High bootstrap support values (≥90) were observed, particularly in the branching patterns distinguishing 'Candidatus P. auctus' nov. sp. JDE115 from its closest relatives, reinforcing its phylogenetic distinctiveness (Fig 3). The closest relatives of JDE115 in the phylogenetic tree include *P. glycinae* MS586 (TYGS ID: 37260), *P. kribbensis* KCTC 32541$^T$ (TYGS ID: 14671), and *P. moraviensis* LMG 24280 (TYGS ID: 19005). The accession numbers and strain names of all related species are summarized in Table 4. These data underscore the unique taxonomic position of JDE115 within the genus *Pseudomonas* based on whole-genome analysis. Having established the genomic distinctiveness of JDE115, its chemotaxonomic characteristics were studied to further validate its taxonomic classification.

**Table 3. ANI (%) and dDDH (%) between 'Candidatus Pseudomonas auctus' nov. sp. JDE115 and closely related sequenced strains of the genus Pseudomonas.**

| Pseudomonas species | Genome accession number | ANI (%) | DDH (%) |
|---|---|---|---|
| P. glycinae MS586 | GCF_001594225.2 | 94.59 | 57.10 |
| P. fitomaticsae FIT81 | GCF_021018765.1 | 94.24 | 54.80 |
| P. gozinkensis IzPS32d | GCF_014863585.1 | 93.92 | 54.00 |
| P. allokribbensis IzPS23 | GCF_014863605.1 | 91.74 | 43.00 |
| P. kribbensis 46−2 | GCF_003352185.1 | 91.46 | 42.50 |
| P. monsensis PGSB 8459 | GCF_014268495.2 | 88.43 | 33.60 |
| P. zeae OE 48.2 | GCF_014268485.2 | 88.20 | 33.00 |
| P. tensinigenes ZA 5.3 | GCF_014268445.2 | 88.12 | 32.90 |
| P. germanica FIT28 | GCF_019614655.1 | 88.07 | 32.70 |
| P. iranensis SWRI54 | GCF_014268585.2 | 87.67 | 32.10 |
| P. hamedanensis SWRI165 | GCF_014268595.2 | 87.32 | 31.70 |
| P. grandcourensis DGS24 | GCF_039909015.1 | 86.53 | 29.60 |
| P. hormoni G20-18 | GCF_018502625.1 | 86.41 | 29.60 |
| P. azerbaijanorientalis SWRI123 | GCF_019139795.1 | 86.41 | 29.90 |
| P. nunensis ln5 | GCF_024296925.1 | 86.29 | 29.30 |
| P. wuhanensis FP607 | GCF_030687395.1 | 86.15 | 29.30 |
| P. svalbardensis PMCC 200367 | GCF_030053115.1 | 86.10 | 29.00 |
| P. cucumis FP1935 | GCF_030687935.1 | 86.07 | 28.90 |
| P. silesiensis | GCF_001661075.1 | 85.76 | 29.30 |
| P. purpurea DGS26 | GCF_039908635.1 | 85.30 | 27.20 |
| P. chlororaphis ATCC 9446 | GCF_036689615.1 | 85.12 | 26.80 |
| P. helvetica DGS28 | GCF_039908645.1 | 85.01 | 27.00 |
| P. ogarae SWRI108 | GCF_014268695.2 | 84.60 | 26.90 |
| P. viciae 11K1 | GCF_004786035.1 | 84.53 | 26.50 |
| P. alvandae SWRI17 | GCF_019141525.1 | 84.29 | 26.60 |
| P. protegens CHA0 | GCF_900560965.1 | 84.16 | 25.40 |
| P. hefeiensis FP205 | GCF_030687835.1 | 84.03 | 26.10 |
| P. shahriarae SWRI52 | GCF_014268455.2 | 83.69 | 25.10 |
| P. pergaminensis 1008 | GCF_024112395.2 | 83.64 | 24.90 |
| P. asgharzadehiana SWRI132 | GCF_019139815.1 | 83.53 | 24.60 |
| P. tritici SWRI145 | GCF_014268275.3 | 83.42 | 24.50 |
| P. salmasensis SWRI126 | GCF_014268375.2 | 83.33 | 24.80 |
| P. vanderleydeniana RW8P3 | GCF_014268755.2 | 83.30 | 24.80 |
| P. sessilinigenes CMR12a | GCF_003850565.1 | 83.07 | 24.30 |
| P. mucidolens NCTC8068 | GCF_900475945.1 | 82.91 | 24.40 |
| P. tructae SNU WT1 | GCF_004214895.1 | 82.11 | 22.90 |
| P. abieticivorans PIA16 | GCF_023509015.1 | 81.89 | 23.10 |
| P. oryziphila 1257 | GCF_003940825.1 | 81.76 | 22.70 |
| P. muyukensis COW39 | GCF_019139535.1 | 81.73 | 22.40 |
| P. eucalypticola NP-1 | GCF_013374995.1 | 81.63 | 22.80 |
| P. fakonensis COW40 | GCF_019139895.1 | 81.63 | 22.40 |
| P. versuta L10.10 | GCF_001294575.1 | 81.62 | 23.30 |
| P. maumuensis COW77 | GCF_019139675.1 | 81.58 | 22.10 |
| P. xantholysinigenes RW9S1A | GCF_014268885.2 | 81.56 | 22.40 |
| P. xanthosomatis COR54 | GCF_019139835.1 | 81.56 | 22.60 |

*(Continued)*

| *Pseudomonas* species | Genome accession number | ANI (%) | DDH (%) |
|---|---|---|---|
| *P. entomophila* L48 | GCF_000026105.1 | 81.52 | 22.40 |
| *P. lijiangensis* LJ2 | GCF_018968705.1 | 81.44 | 22.60 |
| *P. taetrolens* NCTC10697 | GCF_900475285.1 | 81.34 | 23.10 |
| *P. cichorii* DSM 50259 | GCF_018343775.1 | 81.31 | 22.60 |
| *P. anuradhapurensis* RD8MR3 | GCF_014269225.2 | 81.24 | 22.70 |
| *P. rhizosphaerae* DSM 16299 | GCF_000761155.1 | 81.24 | 22.80 |
| *P. putida* NBRC 14164 | GCF_000412675.1 | 81.24 | 22.20 |
| *P. fortuita* GMI12077 | GCF_026898135.2 | 81.05 | 22.10 |
| *P. syringae* pv. tomato str. DC3000 | GCF_000007805.1 | 80.87 | 22.50 |
| *P. cavernae* K2W31S-8 | GCF_003595175.1 | 80.72 | 21.90 |
| *P. huanghezhanensis* BSw22131 | GCF_026810445.1 | 80.40 | 22.00 |
| *P. promysalinigenes* RW10S1 | GCF_014269025.2 | 80.36 | 21.60 |
| *P. syringae* pv. *Tagetis* ICMP 4091 | GCF_022557255.1 | 80.32 | 22.00 |
| *P. tohonis* TUM18999 | GCF_012767755.2 | 80.18 | 21.30 |
| *P. wenzhouensis* A20 | GCF_021029445.1 | 80.10 | 21.60 |

**Tree scale: 0.01** ⊢———⊣

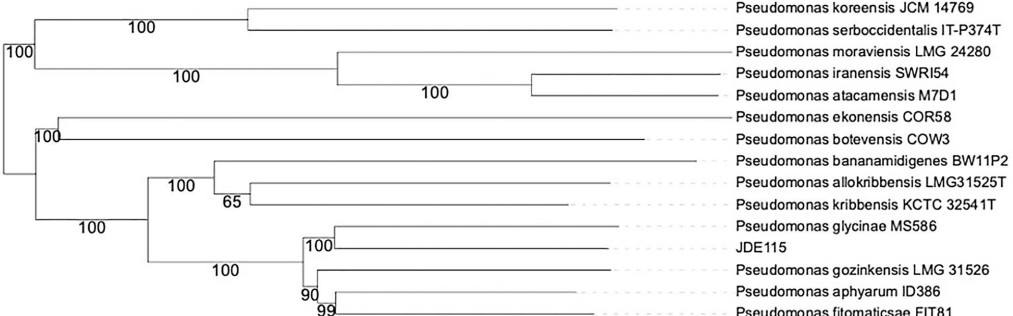

**Fig 3. Whole-genome sequence tree generated with TYGS for 'Candidatus Pseudomonas auctus' nov. sp. JDE115 and its closely related species of the genus *Pseudomonas*.** Tree inferred with FastME from GBDP distances was calculated from genome sequences. Branch lengths are scaled in terms of GBDP distance formula d5; numbers above branches are GBDP pseudo-bootstrap support values from 100 replications. Accession numbers of sequences used in this study are summarized in Table 4.

## Chemotaxonomic analysis

Cellular fatty acids were analyzed for '*Candidatus* P. auctus' nov. sp. JDE115 using the Sherlock 6.1 system (Microbial Identification Inc.) and the RTSBA6 library [67]. The dominant fatty acids detected for JDE115 included C16:0 (25.07%), summed feature 3 (C16:1ω7c/C16:1ω6c) (34.78%), summed feature 8 (C18:1ω7c/C18:1ω6c) (16.91%), and C17:0 cyclo (0.87%). Additionally, fatty acids C12:0 (1.88%), C14:0 (0.66%), and C18:0 (1.35%) was identified (Table 5). These profiles are consistent with those typically observed in the genus *Pseudomonas*. The presence of three characteristic fatty acids—C10:0 3-OH, C12:0, and C12:0 3-OH—further confirms the taxonomic affiliation of JDE115 within *Pseudomonas*.

**Table 4. Accession numbers for genome sequences of various *Pseudomonas* species strains utilized in whole-genome phylogenetic analysis.**

| Species | Biosample accession | Strains name | TYGS ID | Isolation Source | Authority |
|---|---|---|---|---|---|
| *P. bananamidigenes* | SAMN04444410 | BW11P2 | 100258 | Rhizosphere | [122] |
| *P. aphyarum* | SAMN28093775 | ID386 | 104585 | Aquatic environments | [124] |
| *P. fitomaticsae* | SAMN19241968 | FIT81 | 109195 | Rhizosphere | [121] |
| *P. kribbensis* | SAMN09244908 | 32541ᵀ | 14671 | Soil or environmental | [110] |
| *P. iranensis* | SAMN15248341 | SWRI54 | 147537 | Rhizosphere | [122] |
| *P. ekonensis* | SAMN19473669 | COR58 | 147556 | Rhizosphere | [122] |
| *P. moraviensis* | SAMN04490196 | LMG 24280 | 19005 | **Unknown** | [112] |
| *P. atacamensis* | SAMN11356701 | M7D1 | 22501 | Rhizosphere | [116] |
| *P. glycinae* | SAMN04435621 | MS586 | 37260 | Rhizosphere | [73] |
| *P. koreensis* | SAMD00245249 | JCM 14769 | 48496 | Soil or environmental | [130] |
| *P. botevensis* | SAMN19473668 | COW3 | 82604 | Rhizosphere | [122] |
| *P. serboccidentalis* | SAMN29793685 | IT-P374ᵀ | 86200 | Rhizosphere | [125] |
| *P. gozinkensis* | SAMN16250066 | LMG 31526 | 92631 | Soil or environmental | [131] |
| *P. allokribbensis* | SAMN16250065 | LMG31525ᵀ | 92633 | Soil or environmental | [131] |
| '*Candidatus* Pseudomonas auctus' nov. sp. | SAMN45708326 | JDE115 | U884796 | Rhizosphere | This study |

This table provides the genome accession numbers, strain names, TYGS ID, and isolation sources of *Pseudomonas* species used in the whole-genome sequence-based phylogenetic analysis. '*Candidatus* Pseudomonas auctus' nov. sp. JDE115 was sequenced in this study.

**Table 5. Distinct cellular fatty acid profiles of strain '*Candidatus* Pseudomonas auctus' nov. sp. JDE115 compared with related *Pseudomonas* strains.**

| Fatty acid | *P. auctus* JDE115 | *P. glycinae* MS586ᵀ | *P. kribbensis* 46-2ᵀ | *P. granadensis* F-278 | *P. moraviensis* 1B4ᵀ | *P. koreensis* Ps9-14ᵀ | *P.baetica* a390T | *P. vancouverensis* DhA-51ᵀ | *P. jessenii* DSM 17150ᵀ | *P. reinekei* MT1ᵀ |
|---|---|---|---|---|---|---|---|---|---|---|
| C10:0 3-OH | 5.5 | 6.6 | 5.4 | 3.2 | 2.6 | 2.2 | 3.4 | 4.8 | 2.8 | 3.3 |
| C12:0 2-OH | 6.38 | 5.5 | 6.8 | 4.7 | 4.9 | 5 | 5.5 | 3.8 | 5.5 | 4.3 |
| C12:0 3-OH | 5.62 | 6.7 | 7.5 | 2.5 | 4.1 | 4 | 3.2 | 5.7 | 3.2 | 4.8 |
| C10:0 | ND | 0.8 | ND | ND | ND | ND | 0.1 | 0.3 | 0.1 | ND |
| C12:0 | 1.88 | 2.9 | ND | 1.5 | 2.1 | 1.6 | 1.7 | 3.8 | 4.7 | 3.6 |
| C14:0 | 0.66 | 0.6 | 1.2 | ND | 0.4 | 0.7 | 0.5 | 0.6 | 0.3 | 0.7 |
| C16:0 | 25.07 | 22.9 | 33.4 | 32 | 29 | 33 | 29.4 | 29.4 | 29.4 | 36.5 |
| C17:0 cyclo | 0.87 | 10.3 | 15.1 | 6.9 | 2.4 | 2 | 3.2 | 9.4 | 0.9 | 22.3 |
| C18:0 | 1.35 | 0.3 | 1.6 | ND | 0.5 | 0.7 | 0.3 | 0.2 | 0.7 | 0.8 |
| C19:0 ω8c | ND | 1.2 | ND | ND | 0.2 | ND | ND | ND | ND | 0.7 |
| Summed feature 3 | 34.78 | 23.6 | 16.8 | 36 | 36 | 37 | 39.5 | 30.8 | 38.1 | 28 |
| Summed feature 8 | 16.91 | 13.4 | 8.9 | 12 | 17 | 13 | 12.2 | 8.5 | 17.2 | 8.6 |

Fatty acid composition of '*Candidatus* Pseudomonas auctus' nov. sp. JDE115 and closely related *Pseudomonas* species, analyzed using the Sherlock 6.1 system (Microbial Identification Inc.) and the RTSBA6 library. The values represent the percentage of total fatty acids detected. Summed features refer to groups of two or three fatty acids that cannot be individually resolved using gas chromatography (GC) with the MIDI system. Summed feature 3 includes C16:1ω7c and/or C16:1ω6c, while summed feature 8 comprises C18:1ω7c and/or C18:1ω6c. Data for strain '*Candidatus* Pseudomonas auctus' nov. sp. JDE115 were obtained in this study. Data for other type strains were obtained from references, *P. glycinae* MS586ᵀ [73]; *P. kribbensis* 46-2ᵀ [110]; *P. granadensis* F-278,770ᵀ [111]; *P. moraviensis* 1B4ᵀ [111]; *P. koreensis* Ps9-14ᵀ [112]; *P.baetica* a390ᵀ [113]; *P. vancouverensis* DhA-51ᵀ [114]; *P. jessenii* DSM 17150ᵀ [114]; *P. reinekei* MT1ᵀ [114].

Abbreviation: ND indicates fatty acids that were not detected.

Compared to closely related species, such as *P. glycinae* MS586T, which shows C16:0 at 22.9%, summed feature 3 at 23.57%, summed feature 8 at 13.37%, and C17:0 cyclo at 10.28%, JDE115 exhibits a higher proportion of summed features 3 and 8 but lower C17:0 cyclo. These differences emphasize the unique chemotaxonomic profile of JDE115 within the genus. Moreover, the characteristic fatty acids C10:0 3-OH, C12:0, and C12:0 3-OH, commonly observed in *Pseudomonas* species, were detected in '*Candidatus* P. auctus' nov. sp. JDE115, further validating its classification within the genus.

Strains like *P. kribbensis* 46-2$^T$ and *Pseudomonas moraviensis* 1B4$^T$ exhibit variations in their fatty acid profiles. For instance, *P. kribbensis* 46-2$^T$ has a significantly higher proportion of C16:0 (33.4%) and C17:0 cyclo (15.1%) than JDE115, while *P. moraviensis* 1B4$^T$ shows C16:0 at 32% and summed feature 3 at 36%, reflecting its distinct metabolic adaptations (Table 5).

The chemotaxonomic differentiation is further highlighted by unique percentages in summed feature 8 and the presence of minor fatty acids like C19:0 ω8c in JDE115, which were not detected in many other *Pseudomonas* strains. These comparative findings underscore the novelty of '*Candidatus* P. auctus' nov. sp. JDE115's chemotaxonomic profile and its distinct position within the genus.

## Conclusions

Based on a comprehensive polyphasic approach integrating phenotypic, molecular, and chemotaxonomic analyses, '*Candidatus* P. auctus' nov. sp. JDE115 is proposed as a novel species within the genus *Pseudomonas*. Its distinct taxonomic position is strongly supported by divergence in genomic similarity indices, including ANI and GGDC, which fall below the species delineation thresholds compared to its closest relatives. Phenotypic characteristics, such as motility, Gram-negative and rod-shaped morphology, facultative aerobiosis, and catalase- and oxidase-positive activity, align with traits typical of the genus while demonstrating unique growth conditions and fluorescence properties. Additionally, the chemotaxonomic profile, including dominant fatty acids such as C16:0, summed feature 3 (C16:1ω7c/C16:1ω6c), and summed feature 8 (C18:1ω7c/C18:1ω6c), further confirms its placement within *Pseudomonas* while distinguishing it from closely related species. These findings underscore the ecological and taxonomic significance of JDE115, providing a strong foundation for exploring its potential in plant growth promotion and biocontrol. The name '*Candidatus* P. auctus' nov. sp. is proposed. It remains *candidatus* because the type strain cannot be released to the public because of its potential development as a biopesticide.

### Description of '*Candidatus* Pseudomonas auctus' nov. sp.

'*Candidatus* P. auctus' nov. sp. (auc'tus. L. masc. adj. auctus, meaning growth or increase, referring to its ability to promote plant growth) is an aerobic or facultative anaerobic, gram-negative, rod-shaped bacterium with motility conferred by polar flagella. Colonies are fluorescent and appear light-yellow when grown on King's B agar plates. On LB agar, colonies are smooth, circular, slightly convex, and 5.0–7.5 mm in diameter after 72 hours of incubation at 28°C. Cells are approximately 0.7-.09 X 2.2–3.0 µm in size. Growth occurs at temperatures ranging from 4°C to 40°C, with an optimum temperature of 28–30°C. The bacterium can grow within a pH range of 4.0–10.0, with an optimum pH of 6.0–7.0. '*Candidatus* P. auctus' nov. sp. tolerates salinity up to 4% (w/v) NaCl, with optimal growth observed at 1% NaCl.

Biochemical and metabolic characterization using Biolog GENIII Micro Plates (S1 Table) demonstrated the ability of '*Candidatus* P. auctus' nov. sp. JDE115 to utilize a variety of substrates. The strain exhibited positive utilization for several carbohydrates, including α-d-glucose, d-mannose, d-fructose, d-galactose, d-fucose, l-fucose, l-rhamnose, and d-mannitol. Amino acid utilization included d-aspartic acid, l-aspartic acid, l-glutamic acid, l-histidine, l-alanine, l-arginine, l-serine, and glycyl-l-proline. The strain also metabolized organic acids such as citric acid, l-malic acid, α-ketoglutaric acid, γ-aminobutyric acid, d-saccharic acid, and acetic acid. '*Candidatus* P. auctus' nov. sp. JDE115 demonstrated growth in the presence of chemical inhibitors such as fusidic acid, troleandomycin, rifamycin SV, vancomycin, guanidine HCl, and

potassium tellurite, reflecting its adaptive tolerance to diverse chemical conditions. It also exhibited positive results for pH 6 and 5 and tolerated NaCl concentrations up to 4%, but growth was inhibited at 8%. The strain was resistant to aztreonam, nalidixic acid, and lithium chloride but was inhibited by sodium butyrate.

Phylogenetic analysis based on 16S rRNA gene sequences revealed that 'Candidatus P. auctus' nov. sp. JDE115 shares high sequence similarity (>99%) with *P. glycinae* MS586$^T$, *P. kribbensis* 46-2$^T$, and *P. soyarea* JJL17$^T$. However, whole-genome-based ANI (94.59%) and GGDC (57.10%) values with its closest relative, *P. glycinae* MS586$^T$, fall below the thresholds for species delineation, confirming its status as a novel species. The strain JDE115 (=SAMN45708326), isolated from soybean root nodules at Kentland Farm, McCoy, Virginia, USA.

## Supporting information

**S1 Fig. Comparative fluorescence of JDE115 and *Klebsiella* sp.** This image highlights the fluorescence capability of 'Candidatus Pseudomonas auctus' nov. sp. JDE115, evident in three fluorescent plates, compared to the non-fluorescent Klebsiella sp. The test confirms the fluorescence property of JDE115 under UV light.
(TIF)

**S1 Movie. Live observation of 'Candidatus Pseudomonas auctus' nov. sp. JDE115 demonstrating active motility.** This movie was recorded at 630x using phase contrast, causing the bacteria to appear black; the lighting was gradually changed to bright field to make the bacteria appear white and the clear polysaccharide capsule to become visible.
(MP4)

**S2 Fig. Morphological characterization of 'Candidatus Pseudomonas auctus' JDE115 colonies.** The diameters of the colonies were measured with a digital caliper after incubation at 28°C on Luria-Bertani agar for 24hr.
(TIF)

**S1 Table. The physiological, morphological, and phenotypic characteristics of 'Candidatus Pseudomonas auctus' sp. nov. strain JDE115 in Biology GEN III tests.** The tests were repeated three times. This analysis also demonstrated the ability of JDE115 to thrive under aerobic conditions that was evident from the tetrazolium dye reduction assay which functions as an indicator of active respiration. In the BIOLOG system, a positive reaction occurs when the bacterium can metabolize a given substrate, leading to dye reduction and a color change from colorless to purple. The widespread positive metabolic activity across multiple carbon sources confirms the presence of an active electron transport chain (ETC), a hallmark of aerobic respiration. Furthermore, the test results indicated the ability of JDE115 to utilize a broad spectrum of organic acids (e.g., L-malic acid, D-malic acid, citric acid, propionic acid, and acetic acid), which are known to be key carbon sources for aerobic bacteria and facultative anaerobes. This metabolic flexibility indicated a bacterium that efficiently engages in oxidative phosphorylation while still having the capacity to switch to fermentation or alternative electron acceptors when oxygen is limited. The survival of JDE115 under varying NaCl and pH conditions also supports its classification as a facultative aerobe. Its tolerance to 1%, 4%, and 8% NaCl suggests that it can withstand osmotic fluctuations, an important trait for bacteria adapting to different environmental conditions, including those found inside plant tissues.
(DOCX)

**S1 File. Additional references.**
(DOCX)

## Acknowledgments

The authors would like to acknowledge Virginia Tech Intellectual Property for supporting this research and extend our gratitude to the School of Plant and Environmental Sciences at Virginia Tech for providing the resources and facilities necessary to conduct this study.

## Author contributions

**Conceptualization:** Md Sahadat Ali.

**Data curation:** Md Sahadat Ali, David Haak, Paulo Vieira.

**Formal analysis:** Jonathan D. Eisenback, Md Sahadat Ali, Michael Evans, Steven Rideout, David Haak, Paulo Vieira.

**Funding acquisition:** Jonathan D. Eisenback, Md Sahadat Ali.

**Investigation:** Jonathan D. Eisenback, Md Sahadat Ali.

**Methodology:** Jonathan D. Eisenback, Md Sahadat Ali, Fatima Tuz Zohora Mony, Michael Evans, Steven Rideout, Paulo Vieira.

**Project administration:** Jonathan D. Eisenback, Md Sahadat Ali.

**Resources:** Jonathan D. Eisenback, Md Sahadat Ali, Steven Rideout, David Haak.

**Software:** Jonathan D. Eisenback, Md Sahadat Ali, David Haak.

**Supervision:** Jonathan D. Eisenback, Md Sahadat Ali, Michael Evans, Steven Rideout, David Haak, Paulo Vieira.

**Validation:** Jonathan D. Eisenback, Md Sahadat Ali, Fatima Tuz Zohora Mony.

**Visualization:** Jonathan D. Eisenback, Md Sahadat Ali, Fatima Tuz Zohora Mony.

**Writing – original draft:** Jonathan D. Eisenback, Md Sahadat Ali.

**Writing – review & editing:** Jonathan D. Eisenback, Md Sahadat Ali, Steven Rideout, David Haak, Paulo Vieira.

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
