## [Decision Letter · Decision Letter 0]

19 Aug 2025

PONE-D-25-28050‘Candidatus Pseudomonas auctus’ sp. nov. JDE115 isolated from nodules on soybean (Glycines max)PLOS ONE

Dear Dr. Eisenback,

Thank you for submitting your manuscript to PLOS ONE. After careful consideration, we feel that it has merit but does not fully meet PLOS ONE’s publication criteria as it currently stands. Therefore, we invite you to submit a revised version of the manuscript that addresses the points raised during the review process. Please consider the comments done by the reviewers

We look forward to receiving your revised manuscript.

Kind regards,

Guadalupe Virginia Nevárez-Moorillón, Ph.D.

Academic Editor

PLOS ONE

Journal Requirements:

Virginia Tech Intellectual Property LAUNCH Proof-of-Concept Grant

The School of Plant and Environmental Sciences, Virginia Tech research associate and tuition

3. Please remove all personal information, ensure that the data shared are in accordance with participant consent, and re-upload a fully anonymized data set.

Additional guidance on preparing raw data for publication can be found in our Data Policy (https://journals.plos.org/plosone/s/data-availability#loc-human-research-participant-data-and-other-sensitive-data) and in the following article: http://www.bmj.com/content/340/bmj.c181.long .

Reviewers' comments:

Reviewer's Responses to Questions

**Comments to the Author**

1. Is the manuscript technically sound, and do the data support the conclusions?

Reviewer #1: Yes

Reviewer #2: Yes

2. Has the statistical analysis been performed appropriately and rigorously? 

Reviewer #1: Yes

Reviewer #2: I Don't Know

3. Have the authors made all data underlying the findings in their manuscript fully available?

Reviewer #1: Yes

Reviewer #2: Yes

4. Is the manuscript presented in an intelligible fashion and written in standard English?

Reviewer #1: Yes

Reviewer #2: Yes

5. Review Comments to the Author

Reviewer #1: The article is very well structured. The experiments support the proposal made in the article, clearly demonstrating the proposal of a new species of the genus Pseudomonas. My unique question is why in the table 3 title describe the value as GGDC, but in the table itself, the value is labeled as DDH?"

Reviewer #2: The manuscript demonstrates strong technical and do support for the results and conclusions.

All data (figures, tables, videos and supplementary information) are fully available.

I suggest increasing the resolution of figures 2 and 3.

6. PLOS authors have the option to publish the peer review history of their article (what does this mean? ). If published, this will include your full peer review and any attached files.

**Do you want your identity to be public for this peer review?** For information about this choice, including consent withdrawal, please see our Privacy Policy .

Reviewer #1: No

Reviewer #2: No

---

## [Author Response · Author response to Decision Letter 1]

21 Aug 2025

Thank you for your comments, all of which have been incorporated into our manuscript. The figures with poor resolution in the pdf of this manuscript are improved in the original Tiffs that have been submitted.

---

## [Editor Report · Decision Letter 1]

24 Aug 2025

‘Candidatus Pseudomonas auctus’ sp. nov. JDE115 isolated from nodules on soybean (Glycines max)

PONE-D-25-28050R1

Dear Dr. Eisenback,

We’re pleased to inform you that your manuscript has been judged scientifically suitable for publication and will be formally accepted for publication once it meets all outstanding technical requirements.

Kind regards,

Guadalupe Virginia Nevárez-Moorillón, Ph.D.

Academic Editor

PLOS ONE
---

## [Editor Report · Acceptance letter]

PONE-D-25-28050R1

PLOS ONE

Dear Dr. Eisenback,

I'm pleased to inform you that your manuscript has been deemed suitable for publication in PLOS ONE. Congratulations! Your manuscript is now being handed over to our production team.

Kind regards,

on behalf of

Dr. Guadalupe Virginia Nevárez-Moorillón

Academic Editor

PLOS ONE